# Pharmacoinformatic Investigation of Silymarin as a Potential Inhibitor against *Nemopilema nomurai* Jellyfish Metalloproteinase Toxin-like Protein

**DOI:** 10.3390/ijms24108972

**Published:** 2023-05-18

**Authors:** Ravi Deva Asirvatham, Du Hyeon Hwang, Ramachandran Loganathan Mohan Prakash, Changkeun Kang, Euikyung Kim

**Affiliations:** 1College of Veterinary Medicine, Gyeongsang National University, Jinju 52828, Republic of Korea; devabiochem@gnu.ac.kr (R.D.A.); pooh9922@hanmail.net (D.H.H.); mohanprakash111@gmail.com (R.L.M.P.); ckkang@gnu.ac.kr (C.K.); 2Institute of Animal Medicine, Gyeongsang National University, Jinju 52828, Republic of Korea

**Keywords:** jellyfish, venom, transcriptomics, metalloproteinase, alpha fold, molecular docking, molecular dynamics

## Abstract

Jellyfish stings pose a major threat to swimmers and fishermen worldwide. These creatures have explosive cells containing one large secretory organelle called a nematocyst in their tentacles, which contains venom used to immobilize prey. *Nemopilema nomurai*, a venomous jellyfish belonging to the phylum Cnidaria, produces venom (NnV) comprising various toxins known for their lethal effects on many organisms. Of these toxins, metalloproteinases (which belong to the toxic protease family) play a significant role in local symptoms such as dermatitis and anaphylaxis, as well as systemic reactions such as blood coagulation, disseminated intravascular coagulation, tissue injury, and hemorrhage. Hence, a potential metalloproteinase inhibitor (MPI) could be a promising candidate for reducing the effects of venom toxicity. For this study, we retrieved the Nemopilema nomurai venom metalloproteinase sequence (NnV-MPs) from transcriptome data and modeled its three-dimensional structure using AlphaFold2 in a Google Colab notebook. We employed a pharmacoinformatics approach to screen 39 flavonoids and identify the most potent inhibitor against NnV-MP. Previous studies have demonstrated the efficacy of flavonoids against other animal venoms. Based on our analysis, Silymarin emerged as the top inhibitor through ADMET, docking, and molecular dynamics analyses. In silico simulations provide detailed information on the toxin and ligand binding affinity. Our results demonstrate that Silymarin’s strong inhibitory effect on NnV-MP is driven by hydrophobic affinity and optimal hydrogen bonding. These findings suggest that Silymarin could serve as an effective inhibitor of NnV-MP, potentially reducing the toxicity associated with jellyfish envenomation.

## 1. Introduction

Cnidarians are marine invertebrates recognized for their nematocysts, which include jellyfish, sea anemones, and corals. These secretory organelles, found on the body and tentacles, contain toxic components that discharge upon stimulation, leading to local and systemic reactions, and sometimes even death [1,2]. The National Institutes of Health (NIH) estimates that 150 million people get stung by jellyfish during the summer season. Nemopilema nomurai, a common jellyfish, blooms in the western Pacific Ocean’s coastal waters, primarily in China, Korea, and Japan. Although this jellyfish is less toxic than the box jellyfish, it is still known for its enormous size, with a bell that may grow up to 2 m in diameter and tentacles that can extend up to 8 m [3,4].

The biochemical and toxicological effects of jellyfish venoms have been previously studied, including their hemolytic, insecticidal, cardiovascular, antioxidant, enzymatic, and cytotoxic properties [5,6,7,8,9,10]. The venom contains complex toxic proteases, including metalloproteinases and serine proteinases, which are responsible for cytolytic, neurotoxic, cardiotoxic, and hemolytic effects [9,11,12,13,14,15]. Skin injury is a known fact during jellyfish stings, and some studies propose that metalloproteinase may be one of the possible causes. Transcriptomic and proteomic investigations of scyphozoan jellyfish venom revealed that metalloproteinase plays a major role in pathogenesis [16,17,18,19]. It is believed to be the major reason for pain, swelling, and other symptoms associated with jellyfish envenomation [16,20,21]. Further investigation is necessary to completely comprehend the intricate mechanisms of jellyfish venom and to provide efficient treatments.

Flavonoids are a group of secondary plant metabolites categorized into six groups such as flavanol, flavanone, isoflavone, flavone, flavan-3-ol, and anthocyanin [22]. These compounds have various beneficial effects including antioxidant, hepatoprotective, anti-inflammatory, anti-cancer, and anti-viral activities [23,24]. Some of the previous studies proved that the different families of flavonoids can inhibit the potential snake venom metalloproteinases and thereby reduce the hemorrhage in animal models. For example, gallocatechin, isoquercitrin, pinostrobin, and quercetin-3-O-rhamnoside extracted from different plant species have been shown to inhibit the hemorrhagic activity of viperid and elapid venoms, as well as some isolated SVMPs [25,26,27,28]. Likewise, EGCG (belongs to the classes of flavonoids) inhibiting *Nemopilema nomurai* venom metalloproteinases (NnV-MPs) has been demonstrated [29]. The phenolic nuclei of flavonoids possess the ability to interact with proteins through mechanisms such as hydrogen bonding, hydrophobic interactions, metal chelation, and π–π stacking [30]. Flavonoids have the potential to offer therapeutic benefits by chelating metal ions via coordination with oxophilic transition metals. Jellyfish venom metalloproteinases play a significant role in various toxicities. Studying these enzymes can help us gain a better understanding of their function and potentially develop targeted treatments to mitigate their harmful effects.

Hence, the main goal of this study is to find inhibitory molecules for *Nemopilema nomurai* venom metalloproteinase, by screening the potentially bioactive flavonoids using pharmacoinformatic methods. We have obtained some interesting outcomes using computational three-dimensional structure prediction, high throughput ligand screening, pharmacophore mapping, pharmacokinetic profiling, molecular docking, and molecular dynamics. This in silico work will provide deep insight into the molecular characteristics of flavonoid and their mode of action to neutralize the toxin metalloprotein activity of *Nemopilema nomurai*. This research also aids in the further investigation of the potential usage of these bioactive compounds against jellyfish venom metalloproteinase.

## 2. Results

### 2.1. The Sequencing and De Novo Transcriptome Assembly for Nemopilema nomurai Tentacle

The paired-end sequence was obtained from the NCBI-SRA database (Accession No. SRR875120). The rnaSpades assembly program produced a total of 62,127 unique assembled transcripts, with an N50 value of 2538. The GC percentage of the assembly was found to be 41.23%. Based on BUSCO analysis, the overall completeness of the assembly was 92.9%, as shown in Table 1. The assembly produced a total of 20,892 predicted protein-coding sequences (CDS) through BLAST analysis, of which 11,293 were identified by Diamond.

### 2.2. The Putative Toxin-like Metalloprotein Gene Profile of Nemopilema nomurai Tentacle

The *Nemopilema nomurai* tentacle has a diverse set of twenty-eight toxin-like metalloproteinase coding transcripts identified by de novo assemblies. All metalloproteinase toxins were classified into three major families based on the highest Tox-port (e.g., Uniprot/Swiss-Prot), BLAST hit, and biological functions, including fibrinogenolytic, hemostatic, and hemorrhagic. A summary of annotated metalloproteinase contigs for the *Nemopilema nomurai* tentacle is shown in Table 2. In addition, the NCBI, Tox-prot, and Pfam scores for venom metalloproteinase annotation is shown in Appendix A.

### 2.3. Selection of Target Metalloproteinase

Finally, four metalloproteinases were chosen from the *N. nomurai* tentacle transcripts based on their size and classes. These included NnV-Mlp type1, NnV-Mlp type 2, NnV-Mlp type 3, and NnV-Mlp type 4 with the transcript IDs of NODE_22415_length_2148_cov_32.291084_g6860_i3.p1, NODE_8677_length_4065_cov_15.681613_g1415_i5.p1, NODE_41303_length_745_cov_1.141369_g19525_i0.p1, and NODE_8373_length_4132_cov_652.221237_g1984_i1.p1, respectively. Additional information on metalloproteinase annotation is depicted in Table 2, Appendix A and Appendix A.

### 2.4. The Structure Prediction of NnV-Mlp by Alpha Fold2

AlphaFold will generate four protein structures for one model. In this case, we generated five models for each protein and selected the top-ranked protein based on the pLDDT score and pTM score. Figure 1 shows the three-dimensional structures of metalloproteinases (four types) that were modeled using the Alpha fold2 Colab notebook. The pTM metric ranges from 0 to 1 and is used to evaluate protein structure predictions by producing 3D error measurement. When the pTM value is less than 0.2, the predicted residue patterns are either stochastically assigned with negligible or no correlation to the supposed native structure, or they may represent intrinsically disordered proteins. Conversely, when the pTM value is greater than 0.5, the predictions are generally considered strong enough to make reliable inferences. Additionally, the per-residue confidence score is assessed using the predicted local distance difference test (pLDDT) score, which ranges from 0 to 100. Scores above 90 indicate a high level of confidence, while scores below 50 indicate a low level of confidence. In the figures, high confidence scores are denoted in blue, while low confidence scores are denoted in red. The pLDDT scores for (NnV-Mlp) type 1, type 2, type 3, and type 4 were 72.4, 71.3, 75.4, and 91.7, respectively. Furthermore, the pTM scores for (NnV-Mlp) type 1, type 2, type 3, and type 4 were 0.448, 0.594, 0.654, and 0.877, respectively. The results of the pLDDT analysis showed low confidence in the linker region, suggesting potential flexibility. This conclusion is in qualitative agreement with the true per-residue of lDDT-Cα and pLDDT in the NnV-Mlp types. The vertical gray shading indicates amino acid residues that are missing in the experimental structure, whereas the colored shading denotes the minimum and maximum values over five predictions that are shown in Appendix A. The protein structures were validated using a Ramachandran plot which displays the favorable regions of all residues with the phi and psi angles, depicted in Appendix A.

### 2.5. Active Side Prediction

PrankWeb is an interface to P2Rank, a machine-learning method for predicting ligand binding sites based on the local chemical neighborhood of a ligand. Our proposed pocket scoring approach, PRANK, uses a Random Forests classifier to prioritize putative pockets and significantly improves the accuracy of Fpocket and ConCavity [31]. To enable a standardized and objective comparison of predicted binding sites across various protein models (Default, Default + Conservation, Alphafold, Alphafold + Conservation), we calculated the pocket probability score using a specific formula that takes into account the number of true and false pockets predicted by each model on the HOLO4K calibration dataset. The probability score for a particular raw score x is obtained by dividing the number of true pockets with a raw score of ≤x (Tx) by the sum of Tx and the number of false pockets with a score of ≥x (Fx). By calibrating the pocket scoring approach using the HOLO4K dataset, the resulting probability score ranges from 0 to 1 and accurately reflects the proportion of true binding sites among all predicted sites with a similar raw score for each model. Ultimately, this standardized approach enables a fair comparison and evaluation of the performance of different protein models in predicting binding sites [32]. The prank2web method was adopted to explore the ligand binding sites (amino acid residues) The probability scores for NnV-Mlp types 1, 2, 3, and 4 were obtained as 0.512, 0.583, 0.63, and 0.782, respectively. Detailed information on predicted ligand binding sites is provided in the Appendix A. The binding sites of (NnV-Mlp) type 1 (36-GLN,264-HIS,265-THR, GLY-266,267-TYR,268-TYR,270-LEU,296-HIS,297-GLU,300-HIS,305-HIS,306-HIS,309-HIS,325-LYS); (NnV-Mlp) type 2 (175-LYS,182-ASP,183-TYR,184-ASN,185-GLY,186-GLY,187-GLY,189-GLY,208-GLY,211-PHE,212-SER,215-HIS,216-HIS,219-HIS,224-GLU,225-HIS,240-LYS,241-PRO,242-ASN,243-ALA,338-LYS); (NnV-Mlp) type 3 (124-SER,125-SER,126-SER,127-GLY,128-LEU,129-ALA,131-VAL,155-SER,156-THR,159-HIS,160-HIS,163-HIS,168-SER,169-SER,172-ASP,190-ASP,191-ASP,193-ASP), and (NnV-Mlp) type 4 (114-VAL,115-LEU,116-GLY,117-LEU,118-ALA,120-ILE,166-ILE,167-THR,170-HIS,171-GLU,174-HIS,179-GLN,180-HIS,200-ILE,202-TYR,203-PRO,204-ARG,206-THR,433-VAL,434-ASP,435-ALA,436-GLU,437-GLY,438-PRO,439-LEU) are shown in Figure 2.

### 2.6. Molecular Docking

Based on the results mentioned above, selective compounds were docked with the four types of metalloproteinases (NnV-Mlp), type 1 (proteolytic), type 2 (hemorrhagic), type 3 (hemorrhagic), and type 4 (hemorrhagic), to identify potential inhibitors. The lowest RMSD values for the ligand were selected from all conformations generated during the docking analysis. Table 3 displays the hydrogen and hydrophobic interactions, as well as the docking score of the ligand with the protein complex. To evaluate inhibitory efficacy, thirty-three flavonoid compounds were docked against all four types of NnV-Mlps, according to Lipsinki’s rule, and the results are shown in Appendix A. The top three ligands were selected based on their docking score for hydrogen bonding and hydrophobic interaction from the docking results and are displayed in Table 3.

The docked scores of type 1 NnV-Mlp against Silymarin, Pinobanksin, and Tricetin were obtained as −9.5 kcal/mol, −8 kcal/mol, and −8 kcal/mol, respectively. These compounds have a hydrogen bond interaction with amino acid residues of 36-GLN, 96-ARG, 97- GLN, 296-HIS and 39-LEU, 96-ARG, 267-TYR. The docking images are displayed in Figure 3a. Silymarin, Eriodictyol, and Luteolin are the top three compounds that have the highest docking scores of −9.6, −8.9, and −8.9 kcal/mol, respectively, that have docked against NnV-Mlp type 2. The three compounds’ hydrophobic interactions with amino acids were obtained as 461-THR, 465-SER, 381-ARG, 423-CYS, 425-ASP, 428-MET, 430-PHE, 620-ASN and 383-ARG,386-GLY, 430-PHE, 616-ASP, 620-ASN. The docking images are displayed in Figure 3b. The docking results of NnV-Mlp type 3 exhibit high docked scores with Eriodictyol −8.8 kcal/mol, Apigenin −8.8 kcal/mol, and Silymarin −8.8 kcal/mol. Interestingly, the hydrophobic interactions for Eriodictyol, Apigenin, and Silymarin showed similar amino acids such as 124-SER and 125-THR. The docking images are displayed in Figure 3c. Eriodictyol −9.3 (kcal/mol), Silymarin −9 (kcal/mol), and Quercetin −8.8 (kcal/mol) revealed good docking scores against the NnV-Mlp type 4 protein. Interestingly, the hydrophobic interaction for Eriodictyol, Apigenin, and Silymarin showed similar amino acids, such as 115-LEU and 120-ILE. The docking images are displayed in Figure 3d. All the ligands in the docking images are represented in a different color for better visualization.

From these docking results, by checking the highest docking score and hydrogen bond interaction, seven compounds (Silymarin, Pinobanksin, Tricetin, Eriodictyol, Luteolin, Apigenin, Quercetin) were selected for further analyses, such as their DFT, ADMET, and molecular dynamics. Out of all the compounds, Silymarin hit in all four metalloproteinases, hence it was used for further analyses, such as molecular dynamics and Molecular Mechanics Poisson–Boltzmann Surface Area (MMPBSA).

### 2.7. DFT Calculations

Density functional theory is a computational tool for examining the molecular structures and nature of compounds by determining their electron density [33]. The chemical reactivity parameters were calculated by using the frontier orbitals such as HOMO (highest occupied molecular orbitals) and LUMO (lowest unoccupied molecular orbitals). The electric and optical parameters of the structure in the Frontier molecular orbital theory were computed by using a B3LYP/6–31G (d, p) basis set. A band gap is directly related to molecular reactivity since it represents the energy (Egap = ELUMO – EHOMO). DFT calculation formulae were used from Raftani et al. [34]. The statistics of the DFT-based molecular descriptors for the selected seven compounds are given in Table 4. The results showed that Silymarin (0.16372 eV), Tricetin (0.15419 eV), and Pinobanksin (0.17128 eV) are soft molecules due to their low energy gap values. Tricetin (−0.0744 eV) exhibited the highest HOMO, indicating the highest electron donor, while Apigenin (−0.0638 eV) had the lowest LUMO, indicating the highest electron acceptor. Silymarin (−2061.506039 eV) showed a larger electronegativity value, indicating its inhibitory effect. Eriodictyol (4.295892 eV) had the highest dipole moment followed by Quercetin (4.1933281 eV) and Silymarin (3.991373 eV), which is directly proportional to chemical reactivity. Quercetin (0.07932 eV), Luteolin (0.08467 eV), and Silymarin (0.096935 eV) had lower chemical hardness values, indicating good stability. The optimized and HOMO–LUMO structures of the selected seven ligands are given in Figure 4.

### 2.8. ADMET Prediction

In silico ADMET analysis is a rapid way to identify if a molecule has adequate pharmacokinetics and pharmacodynamic properties. The present study selected seven bioactive compounds from flavonoids to check their ADMET properties. The absorption, distribution, metabolism, excretion, toxicity, and physiochemical properties of the seven selected bioactive compounds are represented in Table 5. Cytochrome P450 was checked as it is responsible for drug metabolism. In this study, all the compounds showed better results, thus the metabolism of these ligands will be easier. In excretion properties, all the selected compounds exhibited less than 3 h of half-life which denotes a short-half life. From the selected compounds, Pinobanksin and Silymarin showed significantly less half-life and a better half-life was exhibited by Silymarin, thus these compounds will easily eliminate from the body. From their toxicity properties, all seven compounds were revealed as non-blockers of hERG and AMES, thus there is no effect on the cardiac side, and they did not cause any genetic damage. Among all these compounds, Silymarin showed better volume distribution properties.

### 2.9. Molecular Dynamics by openMM

Molecular dynamics was used to evaluate the effectiveness of Silymarin as an inhibitor against (NnV-Mlp) type 1 (proteolytic), (NnV-Mlp) type 2 (Hemorrhagic), (NnV-Mlp) type 3 (Hemorrhagic), and (NnV-Mlp) type 4 (Hemorrhagic). Various properties were analyzed using simulation, including the root mean square deviation (RMSD), the radius of gyration (Rg), the root mean square fluctuation (RMSF), the distance of the selected amino acid residue interaction, the polar residue interaction, and the hydrogen bond interaction with a ligand throughout the simulation period. The simulation was performed at the time of 20 ns for the toxin–ligand complex.

The RMSD along the C-alpha backbone was monitored to determine the atomistic variation and stability of the (NnV-Mlp) type 1 native protein and the (NnV-Mlp) type 1 inhibitor complexes during the MD simulation. The RMSD plot showed an initial sharp peak increase and then equilibration was attained at 2.5 ns and remained stable for up to 20 ns at an average of 0.5 angstroms to 1.5 angstroms, as shown in Figure 5a(i,ii). The radius of gyration is calculated by determining the root mean square distances from the center of rotation while considering the varying masses of the proteins. The Rg plot (fig) evaluates the ability conformation and folding of the protein at each time point throughout the simulation, providing an understating of the overall trajectory of the protein. The Rg values of the protein and its corresponding ligand complex were found to be similar, with variations ranging from 2.7 nm to 2.9 nm, and ligand values ranging from 5.3 angstroms to 5.7 angstroms, as observed in the case of (NnV-Mlp) type 1 shown in Figure 5a(iii). The RMSF (root mean square fluctuation) of the Silymarin complex was found to be like that of the (NnV-Mlp) type 1 protein, which suggests that the flexibility of the protein does not change significantly upon ligand binding. This is consistent with the idea that loops and secondary structures have varying degrees of flexibility, as shown in Figure 5a(iv).

The RMSD, Rg, and RMSF were used to evaluate the atomistic variation, stability, conformation, and folding of the (NnV-Mlp) type 2 native protein and its inhibitor complex during an MD simulation. The RMSD plot showed stable equilibration at 5 ns, with average values ranging from 5 to 11.5 angstroms. The Rg plot demonstrated similar values for the protein and its complex, ranging from 3.45 nm to 3.75 nm and the ligand values ranged from 5.1 to 5.6 angstroms. The RMSF of the Silymarin complex was found to be like that of the (NnV-Mlp) type 2 protein, indicating that ligand binding does not significantly change the protein’s flexibility. This is consistent with the idea that different regions of the protein have varying degrees of flexibility, as shown in Figure 5b(i–iv).

To evaluate the stability and conformational changes of the natural (NnV-Mlp) type 3 protein and its inhibitor complex during an MD simulation, the RMSD, Rg, and RMSF were examined. The first abrupt shift was observed between 0 and 7 ns with values ranging from 5.0 to 7.5 angstroms, followed by the second occurring at 20 ns, with values from 7.5 to 13.5 angstroms, according to the RMSD plot. Similar Rg values were found for the protein and its complex, with ranges between 2.1 nm and 2.4 nm and ligand values from 5.4 to 5.8 angstroms. The RMSF of the complex was discovered to be comparable to that of the original protein, showing that protein flexibility after ligand binding was not significantly altered. The figure is shown in Figure 5b(i–iv).

During an MD simulation, the (NnV-Mlp) type 4 native protein and its inhibitor complex’s atomistic variation, stability, conformation, and folding were assessed using the RMSD, Rg, and RMSF. According to the RMSD plot, steady equilibration occurred at 4.5 ns, with typical values between 2.0 and 3.5 angstroms, as shown in Figure 5d(i,ii). Similar values for the protein and its complex, ranging from 2.32 nm to 2.4 nm, and ligand values ranging from 5.4 to 5.8 angstroms, were seen on the Rg plot. The (NnV-Mlp) type 4 protein’s RMSF was discovered to be identical to that of the Silymarin complex, suggesting that the flexibility of the protein is not greatly altered by ligand binding. This is shown in Figure 5(iii,iv). The ligand RMSD and the 2D RMSD of ligand figures are depicted in Appendix A.

### 2.10. The Binding Energy between NnV-Mlp Types and Silymarin Using the MMPBSA Method

The molecular dynamic simulation was performed to determine the binding free energy between Silymarin and Nnv-Mlp proteinase using MMPBSA. The interaction energies, such as Van der Waal’s, polar solvation, electrostatic, and binding energy, were calculated for 20 ns of the MD trajectory and are in Table 6. The distances of the ligand interacting with the amino acid residues of the four types of Nnv-Mlp are as follows: type 1—36-GLN, 95-LYS, 96-ARG, 97-GLN, 227-ARG, 230- MET, 267-TYR, 270-LEU, 305-GLU, 306-HIS, 309-LEU; type 2—431-ARG, 452-TYR, 454-LYS, 456- SER, 457-GLY, 458-GLU, 461-THR, 465-SER, 466-SER, 491-ASP,492-CYX, 604-ILE, 608-LYS, 609-PHE; type 3—122-ILE, 123-GLY, 124-SER, 125-THR, 126-VAL, 146-HIS,155-SER,156-THR,159-HIS,190-THR, LE-193, 198-PRO; and type 4—114-VAL, 117-LEU, 118-ALA, 120-ILE, 163-VAL, 166-ILE, 167-THR, 170-HIE, 174-HIE,180-HIE, 202-TYR, 203-PRO, 204-ARG. In addition, the calculations of ligand–amino acid residue distances over 20ns with an average distance of 2.40 ± 0.41, 3.29 ± 0.29, 5.72 ± 1.33, and 3.28 ± 0.79 angstroms of the four types of Nnv-Mlps are shown in Appendix A. The total interaction energies are −56.06 ± 5.17 kcal/mol, −59.78 ± 6.47 kcal/mol, −42.68 ± 7.94 kcal/mol, and −55.48 ± 8.10 kcal/mol. These values were interpreted and calculated from Figure 6.

All the units are given in kcal/mol ΔE (VDWAALS): Van der Waals molecular mechanics energy, ΔE (EEL): Electrostatic molecular mechanics energy, ΔE (EGB): Polar contribution to the solvation energy, ΔE (ESURF): non-polar component of the solvation energy (proportional to the solvent accessible surface area of the solute), ΔG (Gas): Total gas phase molecular mechanics energy, ΔG (Solvation): Total solvation energy, and ΔG (Interaction): Total binding energy.

## 3. Discussion

Jellyfish envenomation has become a global health concern with an estimated 150 million cases reported each year [35]. The symptoms of jellyfish stings can range from severe pain and skin inflammation to dermatitis, nausea, vomiting, and cardiovascular and respiratory distress [36]. Metalloproteases are known for their strong hemorrhagic properties, which include fibrinolysis, apoptosis induction, platelet aggregation inhibition, interference with blood coagulation, inflammation induction, and inactivation of blood serine protease inhibitors [37]. The mechanism of resistance to cnidarian toxin has also been studied through molecular modeling and structural analysis. The metalloprotein from *Nemopilema nomurai* is closely homologous with a snake venom metalloproteinase, as shown in Table 2. In this study, due to the absence of an experimentally proved structure, the three-dimensional structure of a metalloprotein was obtained using the Alphafold2 protein folding approach. AlphaFold uses a physical and geometric inductive bias to build components that learn from the Protein Data Bank (PDB) with few manual characteristics. The protein structure that AlphaFold is trained to generate is the most probable to be present in a PDB structure [38]. The reliability of the modeled protein was assessed by Ramachandran plot by checking the internal energy of the protein molecule and was found to be reasonable.

A total of thirty-nine bioactive compounds were employed from six flavonoid groups and their potential to inhibit NnV-Mpl types of metalloproteinases was analyzed. Flavonoids are a widespread group of polyphenolic compounds that are found in plants and have a benzo-γ-pyrone structure [24]. They have various beneficial properties, including antioxidant, anti-inflammatory, anti-mutagenic, and anti-carcinogenic activities [39]. Numerous flavonoids have been tested on human and animal models and are considered generally safe for use in herbal oral medications [40,41]. Additionally, they effectively inhibit enzymes such as xanthine oxidase, cyclo-oxygenase, protein kinases, and matrix metalloproteinases (MMPs) [42,43,44,45,46]. Compared with standard antioxidant compounds, Silymarin displays strong radical scavenging, hydrogen peroxide scavenging, and metal chelating activity [47]. Metal ions have a chemical affinity that can be utilized by proteins to create enzymes with high catalytic activity. Metal chelation is a widely recognized and commonly used antioxidant method. Antioxidants are known to have strong Fe-binding capabilities due to their functional groups that enable metal binding. The interaction of Fe ions with antioxidant compounds can also modify their biological effects, including their antioxidant properties [48,49].

Evaluating and analyzing hydrogen bonding and hydrophobic interactions can help to select the best compounds, but a thorough analysis is required to determine their preferability over other toxins. Plant-based compounds can undergo molecular interaction studies using pharmacoinformatic applications such as ADMET, molecular docking, and molecular dynamics simulation to deepen our understanding at the molecular and atomic levels. The ADMET properties of compounds are significant in the drug discovery process as they contribute to about 60% of drug failures during various clinical stages. Some of the selected herbal compounds did not pass the ADMET screening. Further studies were conducted to analyze the toxin–ligand interactions (Ligand-Fit) and molecular dynamics results of thirty-nine bioactive compounds with NnV-Mpl metalloproteinase types. These studies were based on the docking score and the stability of interactions between the toxin and ligand complex during 20 ns molecular dynamics simulations, which were used to correlate with the binding affinity.

The inspection of RMSD and RMSF plots showed that the deviations were within acceptable levels and the average residue fluctuations were 0.5 Å–1.0 Å in NnV-Mpl type 4. In NnV-Mpl type 1, NnV-Mpl type 2, and NnV-Mpl type 3, these fluctuations were observed in the protein and protein ligand complexes. The ligand RMSD was stable in NnV-Mpl type 1, NnV-Mpl type 2, and NnV-Mpl type 4, but not in NnV-Mpl type 3. The fluctuations observed in the RMSF plot were primarily in the middle finger region for NnV-Mpl type 1, NnV-Mpl type 2, and NnV-Mpl type 3, but the stability was maintained in NnV-Mpl type 4. No significant changes in overall structural stability were observed upon ligand binding, but the change in conformation of the side chains of the selected residues away from the binding site could significantly affect the binding affinity of the toxin to the receptor, potentially altering the protein–protein interaction site. Herbal compounds have an advantage over other compounds as some of them have already been used in traditional medicine as potential inhibitors for snake venom metalloproteinase.

MD simulations are considered a useful tool for examining the dynamics of protein–ligand interactions. Due to the strong affinity of the ligands, as indicated by the high dock scores and strong molecular interactions, MD simulations and MMPBSA analyses were performed on the best-docked complexes with the inhibitor Silymarin. The results of the MD simulation and MMPBSA studies showed that these compounds were stable inhibitors within the protein binding pocket, and they effectively inhibited the catalytic function of the NnV-Mpl metalloproteinase. These inhibitors have the potential to offer a single therapeutic approach, so further research into structure-based lead optimization is necessary based on these findings regarding the bioactivity of Silymarin.

## 4. Materials and Methods

### 4.1. Bioinformatics Analysis

The *Nemopilema nomurai* tentacle sequence was used in this current study. The species sequence was obtained from NCBI SRA database sequence ID SRR7754710 downloaded using sra-tools (version 2.11.0) [50]. The overall quality of the sequencing run was evaluated using fastqc (version 0.11.9) [51]. Transcriptome assembly was performed using the de novo program rnaSpades (version 3.15.4) [52]. Transcriptome completeness was determined using BUSCO (version 5.3.2) [53]. For the custom annotation pipeline, protein-coding regions were predicted from assembled transcriptome using TransDecoder (version 5.5.0), minimum set to 50 “https://github.com/TransDecoder/TransDecoder (accessed on 24 July 2022)” [54]. Using a blast from NCBI BLAST (version 2.10.1) [55,56], with an e-value cutoff of 0.001, all transcripts were searched against proteins and toxins from the Tox-port animal venom annotation database [57] and all cnidarian toxins and proteins from the protein database on NCBI (cnidaria and ((Toxin or (venom)) were downloaded. Additionally, predicted protein coding regions were searched using hmm search with an e-value cutoff of 0.001 from HMMER 3.1 against hidden Markov model (HMM) profiles from the alignment of venom protein classes. The results from all the searches above (Tox-port, cnidaria-specific NCBI, and hmm search) were combined with only the complete coding sequence used for downstream analysis. The redundant sequence from predicted proteins with a signal peptide was clustered using CD-HIT (version 0.5—2012) with a cutoff of 0.95 [58] and only the top hit from each cluster was used in further analysis. A reciprocal search using blast p was used against the resulting dataset with an e-value cutoff of 1 × 10^−5^ against the Tox-port animal venom database and the NCBI non-redundant protein sequence (nr) database, as well as a hmm-search with an e-value cutoff of 1 × 10^−5^ against Pfam. The result was manually curated to confirm that the blastp annotation from Tox-port matched the detected venom domain from Pfam.

### 4.2. Selection of Metalloproteinase

From the annotation, based on the Tox-prot, NCBI, and Pfam scores, the *N.nomurai* transcript matched with the snake venom metalloproteinase family. Each family chose one *N. nomurai* transcript metalloproteinase based on size and toxic activity.

### 4.3. Preparation of Proteins

The 3D structure of the target proteins prediction was generated in ColabFold using its interface [59]. Input protein sequences were identical to those for AlphaFold modeling. The MMseqs2 method was selected on ColabFold to generate the MSAs, amber relaxation of the model was disabled, and the unpaired MSA prediction was generated. AlphaFold predicts the three-dimensional coordinates of protein residues using pLDDT (predicted lDDT-Cα), which ranges from 0 to 100, and indicates the confidence of each prediction. pLDDT values are stored in the B-factor fields of PDB files and can be extracted using the BioPython library version 1.78. [60].

### 4.4. Selection of Lligands

Research papers published in SCOPUS and PubMed explore were used to screen the ligands. Thirty-nine flavonoids were chosen and screened by Lipinski’s rule of five. Then, thirty-three flavonoid compounds were chosen for further investigation. Commercially available inhibitors were retrieved and chosen for docking against each target. Commercial inhibitors and bioactive compounds were compared to see how commercial drugs interacted with targets in terms of binding affinity and hydrogen bond interactions.

### 4.5. Preparation of Ligands

The selected compounds’ 3D structures were retrieved from the PubChem database [61]. Open Babel software was used to convert the SDF files to PDB format. Each ligand’s angles, torsion roots, charges, and force field were determined and prepared based on the parameters. Finally, the ligand structures were converted to pdbqt format for the docking process.

### 4.6. Active site Prediction

The appropriate active site for each protein must be determined before docking as the ligand will bind to that active site. These active sites will be defined from the ligand’s coordinates in the original target protein retrieved from PDB. Atoms located less than 25 °A were identified as active site residues, allowing the ligand to bind in that site. The active sites of the selected targets were identified by the prank2web server [62].

### 4.7. Molecular Docking

Molecular docking is a computational methodology that uses geometric and scoring analyses to predict the interactions between a protein and small molecules. Autodock Vina is a superior docking software compared with Autodock 4.0, due to its higher mean precision in predicting binding modes, faster speed, and automatic pre-calculation of grid maps that are performed internally [63,64]. PyRx’s AutoDock Vina was used to perform the molecular docking [65,66]. Vina Wizard control was used to select proteins and ligands, and a grid was displayed on proteins. The grid was modified to reflect the active sites of the proteins, which were chosen as better docking sites [63,66,67,68].

### 4.8. Density Functional Theory

Density Functional Theory (DFT) is a computational quantum mechanical modeling tool that correlates calculated molecule energies and is primarily used to provide chemical activity. The B3LYP method and Gaussian 09 6-31G (d, p) basis set were used to calculate the lowest unoccupied molecular orbital (LUMO), highest occupied molecular orbital (HOMO), energy gap, total energy, and chemical potential. Global descriptors responsible for the chemical behavior of molecules such as softness, absolute hardness, electrophilicity index, and electronegativity were also calculated [69].

### 4.9. ADMET Properties

ADMET properties were used to help drug discovery with new targets and compounds with good biological activities. This also predicts physiochemical and pharmacokinetic properties [70]. ADMET Lab server 2.0 tool was used to predict the ADMET properties of the selected ligands. This tool performs multiple drug-likeness analyses and mainly predicts the ADMET-related properties [71].

### 4.10. Molecular Dynamics Simulation

The protein–ligand simulation was carried out on the Google colab framework interface [72,73]. The MD simulations were carried out using the OpenMM Python package [74]. The systems were solvated in an orthorhombic periodic box with OPC four-point water, neutralized by adding Na+ or Cl- ions, and a 12 Å distance was placed between the protein box boundaries [75]. The Amber ff14SB force field was then applied to the systems [76]. The topology and parameter files were prepared using tLeap from the Amber Tools suite (version 19) [77]. In order to identify electrostatic interactions in OpenMM, we utilized Particle Mesh Ewald (PME) [78] and the non-bonded interactions had a cut-off of 10 Å. The preparation protocol consisted of solvent geometry minimization, followed by whole system minimization, then 5 ns equilibration in the NVT ensemble. The system was gradually heated to 300 K during this equilibration while a weak harmonic potential restraint was placed on the solute. Subsequently, equilibration was carried out in the NPT ensemble while maintaining a weak harmonic restraint on the solute until the desired density was achieved. Finally, a 20 ns production simulation was conducted in the NPT ensemble at 1 atm and 300 K. The temperature was kept constant by Langevin integration and pressure regulated by a Monte Carlo barostat [79]. Finally, the MD runs used a 2 fs time step, and 10 ps trajectory snapshots were recorded. The same procedure was used to simulate the Silymarin–NnV-Mpl type metalloproteinase complex; ligand GAFF [80] and ligand parameters were estimated using AM1-BCC [81] level of theory with an antechamber [82]. The MDTraj [83] and Pytraj [84] Python libraries were utilized for trajectory studies; py3Dmol was used for visualization; and Scikit-Learn, SciPy, matplotlib, and NumPy Python libraries were employed for statistical and numerical analysis [85,86,87].

### 4.11. The Prediction of Relative Binding Energy between NnV-Mlp Types and Silymarin Using MMPBSA Method

Twenty nanosecond production runs were utilized for the prediction of relative binding energies calculated utilizing the Molecular Mechanics/Poission–Boltzmann Surface Area. The binding free energy of NnV-Mlp complexes was calculated from 2000 frames extracted at every 10 ps for 20 ns of MD simulation. The per-residue decomposition analysis was undertaken to estimate the qualitative analysis of the energetic contribution of important amino acids of NnV-Mlp with Silymarin. MMPBSA.py [88] of Amber tools was employed for the estimation of relative binding energies and the contribution of amino acids to the calculated binding energy [89,90].

## 5. Conclusions

This study analyzed the interaction between flavonoids and classes of venom metalloproteinase (NnV-Mlp) found in *Nemopilema nomurai* jellyfish. The 3D structure of NnV-Mlp types of metalloproteinases was used to screen various phytochemical compounds, to select the inhibitory efficient candidate. Furthermore, computational methods such as ADMET, molecular docking, DFT, MMPBSA, and molecular dynamics simulations were employed to study the molecular interactions. These findings indicate that the pharmacoinformatic techniques were useful in exploring the interactions between flavonoids against toxins such as NnV-Mlp metalloproteinase. Specifically, this study suggests that silymarin, a plant-based compound, might be used as an inhibitor against metalloproteinases for jellyfish envenomation. Docking algorithms are widely used for their ability to quickly generate poses and scores, but their accuracy is limited by various approximations, such as scoring function accuracy, conformational sampling, ligand flexibility, receptor flexibility, and solvent effects. Despite their limitations, these algorithms remain popular because they are user-friendly. The findings of this study provide a micro-level exploration of the inhibitory activity of Silymarin against NnV-Mlp types of metalloproteinases. However, further in vitro or in vivo studies are necessary to confirm and validate these results. Our future plan is to express the recombinant jellyfish venom metalloproteinase to identify potential inhibitors, such as Silymarin.

## Figures and Tables

**Figure 1 ijms-24-08972-f001:**
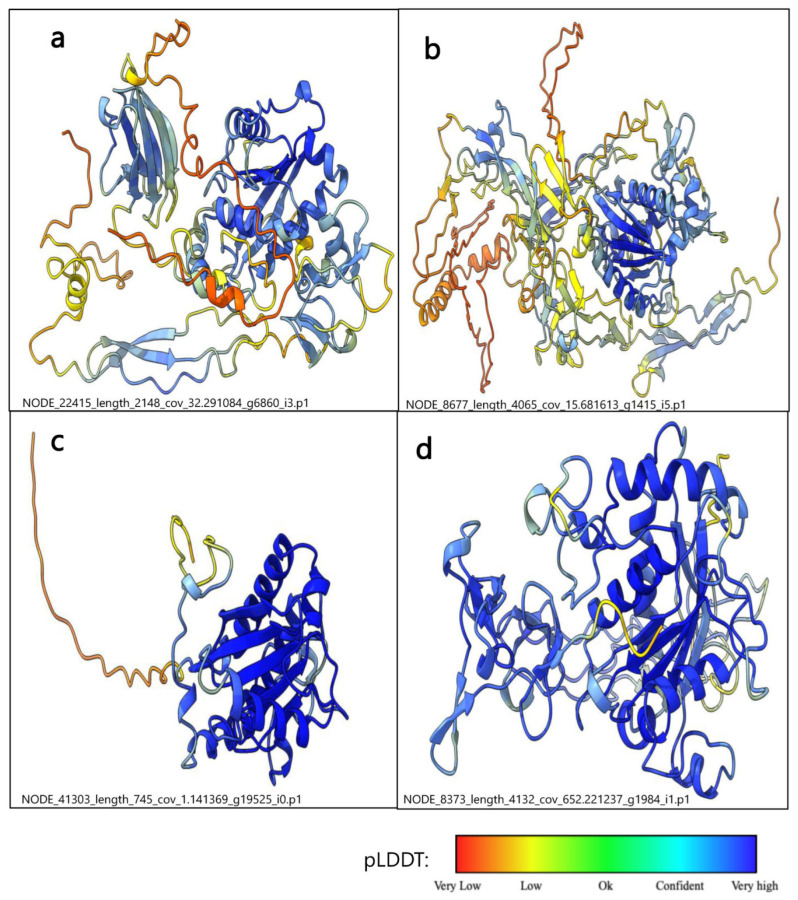
Prediction of the protein complex structures by ColabFold and models were ranked based on AlphaFold pTM score. (**a**) Type 1 NnV-Mlp—NODE_22415_length_2148_cov_32.291084_g6860_i3.p1; (**b**) Type 2 NnV-Mlp—NODE_8677_length_4065_cov_15.681613_g1415_i5.p1; (**c**) Type 3 NnV-Mlp—(NODE_41303_length_745_cov_1.141369_g19525_i0.p1); (**d**) Type 4 NnV-Mlp—(NODE_8373_length_4132_cov_652.221237_g1984_i1.p1).

**Figure 2 ijms-24-08972-f002:**
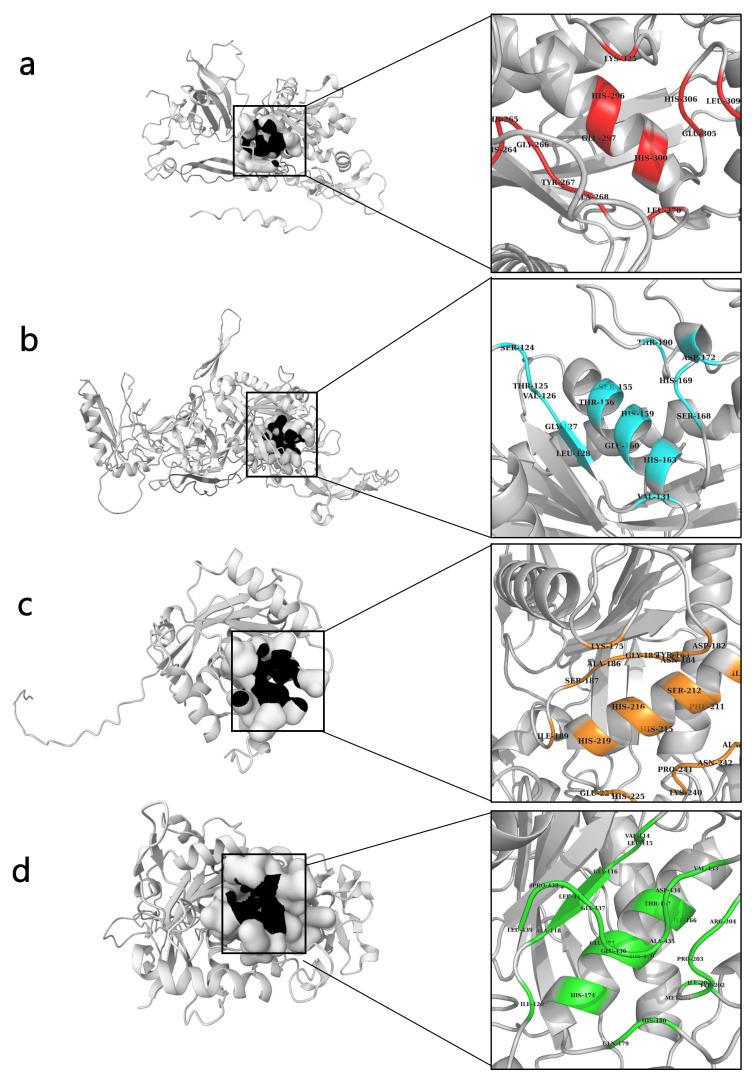
Prediction of ligand binding site (Black color) from prank2web and cartoon representation of binding pocket residues visualized using PyMOL. (**a**) Type 1 NnV-Mlp; (**b**) Type 2 NnV-Mlp; (**c**) Type 3 NnV-Mlp; (**d**) Type 4 NnV-Mlp.

**Figure 3 ijms-24-08972-f003:**
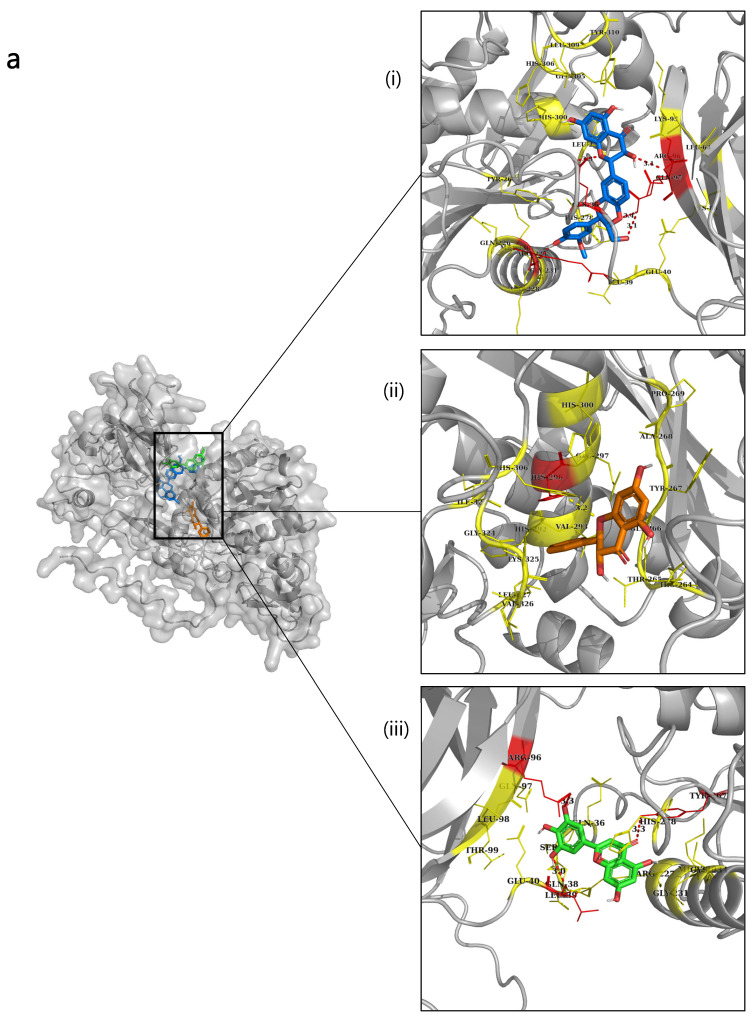
Molecular docking studies of flavonoids against NnV-Mlp types and validated using AutoDock Vina and PyMOL. (**a**) The binding poses of flavonoids against Type 1 NnV-Mlp receptor; (i) Silymarin (blue), (ii) Pinobanksin (orange), and (iii) Tricetin (green); (**b**) The binding poses of flavonoids against Type 2 NnV-Mlp receptor; (i) Silymarin (blue), (ii) Eriodictyol (magenta), and (iii) Luteolin (black); (**c**) The binding poses of flavonoids against Type 3 NnV-Mlp receptor; (i) Silymarin (blue), (ii) Eriodictyol (magenta), and (iii) Apigenin (cyan); (**d**) The binding poses of flavonoids against Type 4 NnV-Mlp receptor (i) Silymarin (blue), (ii) Eriodictyol (magenta), and (iii) Quercetin (purple). The yellow color shows the hydrophobic interaction residues with ligands. The red color shows the hydrogen bond interaction residues with ligands.

**Figure 4 ijms-24-08972-f004:**
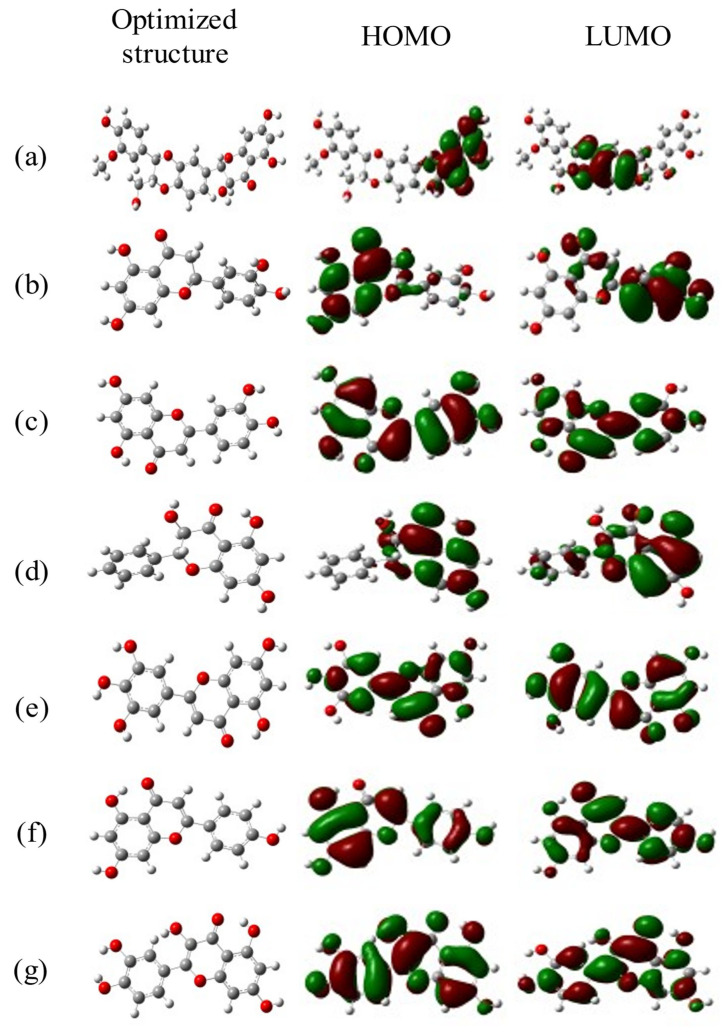
Density Functional Theory (DFT) is a computational quantum mechanical modeling. (**a**) Silymarin, (**b**) Eriodictyol, (**c**) Luteolin, (**d**) Pinobanksin, (**e**) Tricetin, (**f**) Apigenin, and (**g**) Quercetin.

**Figure 5 ijms-24-08972-f005:**
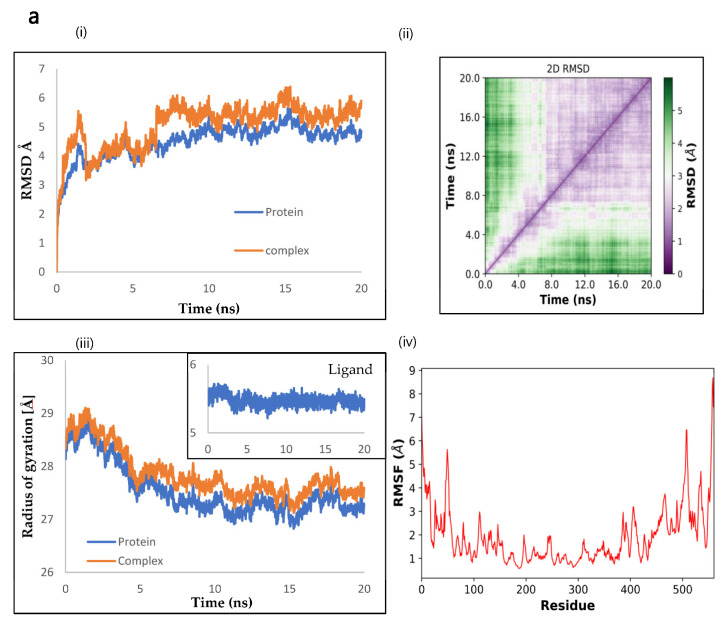
Molecular dynamic studies of Silymarin against NnV-Mlp types, (**a**) NnV-Mlp type 1; (**b**) NnV-Mlp type 2; (**c**) NnV-Mlp type 3; (**d**) NnV-Mlp type 4. (i) overall structural level RMSD were recorded for 2000 frames, (ii) 2D RMSD, (iii) Radius of gyration, (iv) Root Mean Squared Fluctuation (RMSF).

**Figure 6 ijms-24-08972-f006:**
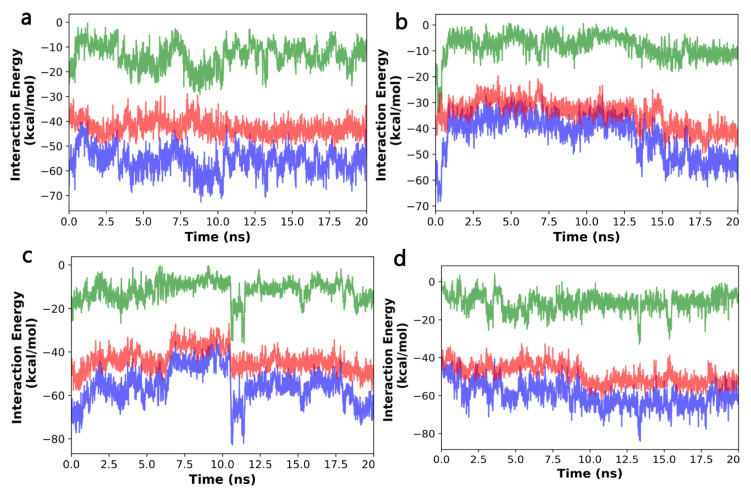
Interaction energy (green color denotes Van der Waals energy; red color denotes electrostatic energy; blue color denotes total energy). (**a**) NnV-Mlp type 1; (**b**) NnV-Mlp type 2; (**c**) NnV-Mlp type 3; (**d**) NnV-Mlp type 4.

**Table 1 ijms-24-08972-t001:** Assembly parameters for the *Nemopilema nomurai* tentacle.

Parts	Reads	Transcripts	Genes	N50	BUSCO%	Diamond Tool
Tentacle	19,627,288	62,127	52,072	2538	92.9	11,293

**Table 2 ijms-24-08972-t002:** Toxin metalloproteinase identified from the *Nemopilema nomurai* tentacle from Tox-port database.

Tox-Prot_Top_Hit	Molecular Function	Class	Length
*Snake venom metalloproteinase adamalysin-2	Metalloendopeptidase activity	P-I	561
*Snake venom metalloproteinase HT-2	Hemostatic and hemorrhagic	P-I	852
Snake venom metalloproteinase	Hemostasis imparing toxin	P-I	863
Snake venom metalloproteinase acutolysin-A	Hemostatic and hemorrhagic	P-I	1569
Snake venom metalloproteinase acutolysin-C	Hemostatic and hemorrhagic	P-I	1543
Snake venom metalloproteinase atrolysin-B	Hemostatic and hemorrhagic	P-I	1482
Snake venom metalloproteinase BITM02A	Hemostatic and hemorrhagic	P-I	1145
*Zinc metalloproteinase-disintegrin bilitoxin-1	Fibrinogenolytic toxin	P-II	240
Zinc metalloproteinase-disintegrin BlatH1	Hemostatic and hemorrhagic	P-II	520
Zinc metalloproteinase-disintegrin BA-5A	Hemostatic and hemorrhagic	P-II	756
Zinc metalloproteinase/disintegrin	Hemostatic and hemorrhagic	P-II	627
Zinc metalloproteinase/disintegrin	Hemostatic and hemorrhagic	P-II	1869
*Zinc metalloproteinase-disintegrin-like 2a	Hemostasis imparing toxin	P-III	439
Zinc metalloproteinase-disintegrin-like 8	Hemostasis imparing toxin	P-III	1130
Zinc metalloproteinase-disintegrin-like ACLD	Toxin activity	P-IIIa	192
Zinc metalloproteinase-disintegrin-like BfMP	Toxin activity	P-IIIa	1131
Zinc metalloproteinase-disintegrin-like BjussuMP-1	Fibrinogenolytic toxin	P-IIIa	1209
Zinc metalloproteinase-disintegrin-like Eoc1	Hemostatic and hemorrhagic	P-IIIc	240
Zinc metalloproteinase-disintegrin-like daborhagin-K	Protease	-	96
Astacin-like metalloprotease toxin 1	Auxillary	-	744
Astacin-like metalloprotease toxin 2	Auxillary	-	548
Astacin-like metalloprotease toxin 3	Auxillary	-	459
Astacin-like metalloprotease toxin 4	Auxillary	-	317

* selected metalloproteinases for further analyses.

**Table 3 ijms-24-08972-t003:** The residue interactions and docking score of the selective bioactive compounds against NnV metalloproteinase.

Modeled Metalloproteinase	Compound Name	Autodock Score (kcal/mol)	Hydrogen Bond Interactions	Hydrophobic Interactions
Type 1 NnV-Mlp	Silymarin	−9.5	36-GLN, 96-ARG, 97-GLN 296-HIS,	39-LEU, 40-GLU, 64-LEU, 88-LYS, 95-LYS, 226-GLN, 228-ARG, 230-MET, 231-GLY, 267-TYR, 270-LEU, 278-HIS, 300-HIS, 305-GLU, 306-HIS, 309-LEU, 310-TYR
	Pinobanksin	−8	296-HIS	264-HIS, 265-THR, 266-GLY,267-TYR,268-ALA,269-PRO,292-HIS,293-VAL,297-GLU,300-HIS,306-HIS
	Tricetin	−8	39-LEU, 96-ARG, 267-TYR	36-GLN,37-SER,38-GLN,40-GLU, 97-GLN, 98-LEU,99-THR, 227-ARG,230-MET, 231-GLY,234-CYS,278-HIS
Type 2 NnV-Mlp	Silymarin	−9.6	461-THR, 465-SER	431-ARG, 454-LYS, 456-SER, 457-GLY, 462-SER,463-MET, 466-SER, 493-PHE, 495-TYR, 608-LYS, 610-ASP, 612-THR
	Eriodictyol	−8.9	381ARG, 423CYS, 425-ASP, 428MET, 430-PHE, 620-ASN	386-GLY,422-PRO,424-PRO, ILE-427, ASN-429, GLN-433, ASP-491,502-LYS,612-THR,613-THR,614-GLY,616-ASP,617-ASP
	Luteolin	−8.9	383ARG, 386GLY, 430-PHE, 616-ASP, 620-ASN	385-CYS,387-GLY,423-CYS,424-PRO,425-ASP,427-ILE,428-MET,429-ASN,433-GLN,491-ASP,502-LYS,612-THR,613-THR,614-GLY,617-HIS,621-THR
Type 3 NnV-Mlp	Eriodictyol	−8.8	125-THR,	124-SER, 126-VAL, GLY-127,128-LEU,129-ALA,155-SER,156-THR,159-HIS,160-GLU,163-HIS,169-HIS,187-CYS, ILE-188,190-THR,191-GLU,192-SER,193-ILE,198-PRO
	Apigenin	−8.8	124-SER, 125-THR	124-SER, 126-VAL, GLY-127,128-LEU,129-ALA,155-SER,156-THR,159-HIS,160-GLU,163-HIS,169-HIS,187-CYS, ILE-188,190-THR,191-GLU,192-SER,193-ILE,198-PRO
	Silymarin	−8.8	124-SER	125-THR, 126-Val, 129-LEU, 131-VAL, 155-SER, 156-THR, 159-HIE, 163-HIE, 169-HIE,190-THR, 191-GLU,192-SER, 193-ILE,198-PRO
Type 4 NnV-Mlp	Eriodictyol	−9.3	170-HIS, 200-ILE	113-GLY, 114-VAL, 115-LEU, 117-LEU, 118-ALA, 163-VAL, 166- ILE, 167-THR, 170-HIS, 171-GLU, 174-HIS, 200-ILE, 201-MET, 202-TYR, 203-PRO, 204-ARG, 206-THR, 423-TYR, 433-VAL, 434-ASP, 437-GLY, 438-PRO
	Silymarin	−9	115-LEU, 120-ILE	114-Val, 117-LEU, 118-ALA, 163-VAL, 166-ILE, 167-THR, 170-HIE, 174-HIE, 180-HIE, 202-TRY, 203-PRO, 204-ARG, 206-THR, 436-GLU, 437-GLY, 438-PRO, 439-LEU.
	Quercetin	−8.8	167-THR, 203-PRO, 434-ASP, 435-ALA, 437-GLY	115-LEU,116-GLY, 117-LEU,118-ALA, 163-VAL, 166-ILE, 170-HIS, 171-GLU,174-HIS,180-HIS, 200-ILE, 202-TYR, 204-ARG, 205-ALA, 206-THR, 423-TYR, 433-VAL, 438-PRO

**Table 4 ijms-24-08972-t004:** Density functional theory analysis to predict the chemical reactive properties.

Compound	Dipole (eV)	HOMO (eV)	LUMO (eV)	Energy Gap (eV)	Chemical Hardness (eV)	Chemical Potential (eV)	Electro-Negativity (eV)
Silymarin	3.991373	−0.05581	−0.21953	0.16372	0.096935	−0.098335	−2061.506039
Eriodictyol	4.295892	−0.05173	−0.22653	0.1748	0.10579	−0.10379	−1031.355218
Luteolin	3.9532421	−0.22785	−0.07296	0.30081	0.08467	−0.12126	−1601.415599
Pinobanksin	3.003841	−0.06836	−0.23964	0.17128	0.09728	−0.10047	−2061.505616
Tricetin	2.455043	−0.0744	−0.22859	0.15419	0.12351	−0.09515	−1248.589238
Apigenin	3.8518676	−0.21666	−0.0638	0.28046	0.10482	−0.09754	−1106.568277
Quercetin	4.1933281	−0.22615	−0.07566	0.30181	0.07932	−0.14842	−1138.910421

**Table 5 ijms-24-08972-t005:** ADMET properties of the selective bioactive compounds.

Properties	Silymarin	Eriodictyol	Luteolin	Pinobanksin	Tricetin	Apigenin	Quercetin
Absorption Properties							
Caco-2 Permeability (Optimal: higher than −5.15 Log unit or −4.70 or −4.80)	−6.255	−5.189	−5.028	−5.193	−5.255	−4.847	−5.204
Human Intestinal Absorption (HIA) ≥ 30%: HIA+; (HIA) <30%: HIA−	-	---	---	---	--	---	---
P-glycoprotein Substrate	---	---	--	---	--	++	---
P-glycoprotein Inhibitor	-	---	---	---	---	---	---
Distribution Properties							
PPB (Plasma Protein Binding): optimal- < 90%	96.66%	93.32%	95.44%	82.08%	92.23%	97.26%	95.50%
VD (Volume Distribution) 0.04–20 L/kg	0.649	0.561	0.533	1.686	0.603	0.51	0.579
Blood–Brain Barrier (BBB) BB ratio ≥ 0.1: BBB+; BB ratio < 0.1:BBB−	---	---	---	---	---	---	---
Metabolism							
CYP1A2 inhibitor	--	++	+++	--	+++	+++	+++
CYP1A2 substrate	--	--	--	--	--	--	--
CYP2C19 inhibitor	--	-	--	---	---	+	---
CYP2C19 substrate	---	---	---	-	---	---	---
CYP2C9 inhibitor	+	++	+	-	+	+	+
CYP2C9 substrate	++	++	++	+++	+	+++	+
CYP2D6 inhibitor	-	+	+	-	--	++	-
CYP2D6 substrate	-	-	+	-	--	++	--
CYP3A4 inhibitor	++	+	+	---	--	++	-
CYP3A4 substrate	-	--	---	-	---	--	---
Excretion							
T 1/2 (Half Lifetime) (long half-life 3 h, short half-life <3 h)	0.274	0.856	0.898	0.68	0.922	0.856	0.929
Toxicity Properties							
hERG (hERG Blockers)	---	---	---	---	--	---	---
AMES (Ames Mutagenicity)	-	+	+	--	-	-	+
DILI (Drug Induced Liver Injury)	+++	+++	+++	+++	+++	++	+++
Physicochemical properties							
LogS (Solubility)	−4.792	−3.827	−3.629	−2.709	−3.59	−3.606	−3.671
LogD (Distribution Coefficient D)	2.524	2.307	2.361	0.688	1.501	2.704	1.767
LogP (Distribution Coefficient P)	2.015	2.118	2.902	1.535	2.52	3.307	2.155

The prediction values are denoted as: (---) = 0–0.1; (--) = 0.1–0.3; (-) = 0.3–0.5; (+) = 0.5–0.7; (++) = 0.7–0.9; (+++) = 0.9–1.0.

**Table 6 ijms-24-08972-t006:** Molecular Mechanics/ Poisson–Boltzmann Surface Area (MMPBSA) binding energy of NnV-Mlp and Silymarin complexes determined for 20ns.

Energy Component (kcal/mol)	NnV-Mlp Type 1	NnV-Mlp Type 2	NnV-Mlp Type 3	NnV-Mlp Type 4
**Δ** **E (VDWAALS)**	−43.7761 ± 1.3006	−48.3463 ± 1.6737	−31.9381 ± 2.4216	−45.4069 ± 1.7482
**Δ** **E (EEL)**	−29.3170 ± 2.6631	−18.8026 ± 1.9482	−23.3120 ± 3.3445	−16.3162 ± 2.1864
**Δ** **E (EGB)**	51.3345 ± 2.4503	44.1952 ± 1.5961	40.2324 ± 3.6828	37.6677 ± 2.1287
**Δ** **E (ESURF)**	−6.7133 ± 0.4014	−7.1048 ± 0.1227	−4.7373 ± 0.2719	−5.9816 ± 0.2048
**Δ** **G (Gas)**	−73.0931 ± 2.4925	−67.1489 ± 2.6451	−55.2500 ± 5.1796	−61.7231 ± 2.8195
**Δ** **G (Solvation)**	83.7113 ± 2.5896	37.0903 ± 1.5079	35.4952 ± 3.4593	31.6860 ± 2.0141
**Δ** **G (Interaction)**	−28.4719 ± 1.7784	−30.0585 ± 1.3585	−19.7549 ± 1.8149	−30.0370 ± 1.2377

## Data Availability

Not applicable.

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
