# Peer review of "Pharmacoinformatic Investigation of Silymarin as a Potential Inhibitor against Nemopilema nomurai Jellyfish Metalloproteinase Toxin-like Protein"

_ijms, 2023, doi:10.3390/ijms24108972_

Round 1

Author Response

Answers to the reviewer's Comment

Q1: Authors didn't provide the details of how many models were generated using Alphafold for each of the metalloproteinase toxin-like proteins.

A1: Thank you for your comments, we have included the information in the revised version of the manuscript. (AlphaFold will generate four protein structures for one model. In this case, we generated five models for each target protein and selected the top-ranked protein structure from each model based on the pLDDT score and pTM score. Hence, total of 20 protein structures were analyzed. From this top 5 protein structures were selected and the detailed information is in Supplementary Figure S1). (Line numbers: 120-122)

Q2) Please compare the binding sites in the four types of proteins and explain their sizes and electrostatic properties.

A2: Thank you for your valuable comment. We have revised our manuscript according to your kind suggestion as “The probability scores for NnV-Mlp type 1, 2, 3, and 4 obtained were 0.512, 0.583, 0.630, and 0.782, respectively. Detailed information on the predicted ligand binding sites was provided in Supplementary Table S2.  The binding sites for each type of NnV-Mlp were as follows;  (NnV-Mlp) type 1: (36-GLN,264-HIS,265-THR, GLY-266,267-TYR,268-TYR,270-LEU,296-HIS,297-GLU,300-HIS,305-HIS,306-HIS,309-HIS,325-LYS); (NnV-Mlp) type 2: (175-LYS,182-ASP,183-TYR,184-ASN,185-GLY,186-GLY,187-GLY,189-GLY,208-GLY,211-PHE,212-SER,215-HIS,216-HIS,219-HIS,224-GLU,225-HIS,240-LYS,241-PRO,242-ASN,243-ALA,338-LYS ); (NnV-Mlp) type 3: (124-SER,125-SER,126-SER,127-GLY,128-LEU,129-ALA,131-VAL,155-SER,156-THR,159-HIS,160-HIS,163-HIS,168-SER,169-SER,172-ASP,190-ASP,191-ASP,193-ASP) and (NnV-Mlp) type 4: (114-VAL,115-LEU,116-GLY,117-LEU,118-ALA,120-ILE,166-ILE,167-THR,170-HIS,171-GLU,174-HIS,179-GLN,180-HIS,200-ILE,202-TYR,203-PRO,204-ARG,206-THR,433-VAL,434-ASP,435-ALA,436-GLU,437-GLY,438-PRO,439-LEU), which were shown in Figure 2.” (Line numbers: 166-178)

Reviewer 2 Report

In this work, the authors performed computational modeling to identify potential nhibitors against Nemopilema nomurai jellyfish metalloproteinase toxin-like protein. Below are my comments:

1. Section 2.4: the authors need to give more information about pLDDT and ptm scores. For example, what is a good score and how are the scores reported in this work related to the confidence of the predicted structures.

2. The resolution of Figure 1 needs to be improved. 

3. Section 2.5: more information about prank2web is necessary. Any previous studies to support this method is reliable? The authors used this method for a critical task in this work but did not give any information to convince readers that the method is robust. 

4. The resolution of Figure 2 needs to be improved. Also the authors need to clarify the confidence level of the structures of these predicted active sites from Alphafold2 (make a connection with Figure 1). If the structures are not reliable then these sites should not be used for further studies. 

5. Section 2.6, line 154: RMSD of what? And what is the reference in the calculation?

6. Section 2.6: Based on the authors' description, it seems the authors considered the docking scores as the binding free energies. This is an incorrect interpretation of docking scores. The docking score can be used for ranking docked poses but does not indicate how tight the compound can bind to the receptor (see https://doi.org/10.1002/cmdc.202200425 and https://doi.org/10.1021/jm050362n). The authors should clarify this in the manuscript and highlight this fact with these previous studies to make sure readers do not mis-use docking scores in their own research. In conclusion, docking is not an ideal tool to predict binding free energies.

7. Table 3,4,5: the authors need to ensure the significant figures are consistent (as well as the numbers reported in the main text).

8. Figure 3: the authors need to color the compound based on the atom type (C,N,O,H ,etc.). Now it is not easy to see what atoms are forming interactions with protein atoms.

9. Section 2.9: I'm a little surprised to see that the authors only performed 20 ns simulations (single trajectory for each system) for such a size system. In Figure 5b it is clear that the simulation is not converged (large RMSD change close to the end of the simulation). Also, the RMSD changes in Figure a,b,c are so large in simulations, making me doubt about the docked poses. If the docked poses are not stable, then longer simulations are needed for more reliable analysis. I suggest the authors perform longer simulations (> 200ns). I also notice the authors used PME for electrostatic interactions in simulations. I suggest the authors can try Reaction Field (RF) since previous studies showed that RF can achieve a similar accuracy as PME while RF is much more efficient than PME (https://pubs.acs.org/doi/abs/10.1021/acs.jcim.0c01424, https://onlinelibrary.wiley.com/doi/10.1002/jcc.20828)). If RF is not available in OpenMM then maybe the authors can mention RF in the manuscript so that more readers are aware of this method and it can be used for either longer or cheaper simulations in future work.

10. Table 6: the authors need to add unit for these numbers

11. Section 2.10: why the authors stated they calculated relative binding energies whereas the absolute binding free energies are reported here (and in Table 6)? Also, since MMPBSA highly depends on the MD simulation. Given the convergence issue I mentioned above, I don't know how much I can trust these numbers. 

12. Since the authors failed to provide any experimental data, it is important to make sure the computational modeling is performed with careful designs and rigorous methods. This is the main reason for my comments above.

The English sometimes is hard to understand. The font is not consistent in the manuscript. 

Author Response

Answers to the reviewer's Comment

Q1. Section 2.4: the authors need to give more information about pTM and pLDDT scores. For example, what is a good score and how are the scores reported in this work related to the confidence of the predicted structures?

A1. Thank you for your valuable suggestion. We have included the details of pTM and pLDDT as 'The pTM metric ranges from 0 to 1 and is used to evaluate protein structure28. predictions by producing 3D error measurements. When the pTM value is less than 0.2, the predicted residue patterns are either stochastically assigned with negligible or no correlation to the supposed native structure, or they may represent intrinsically disordered proteins. Conversely, when the pTM value is greater than 0.5, the predictions are generally considered strong enough to make reliable inferences. Additionally, the per-residue confidence score is assessed using the predicted local distance difference test (pLDDT) score, which ranges from 0 to 100. Scores above 90 indicate a high level of confidence, while scores below 50 indicate a low level of confidence. In the figures, high confidence scores are denoted in blue, while low confidence scores are denoted in red.' (Line numbers: 123-130)

Q2. The resolution of Figure 1 needs to be improved. 

A2. Thank you for your suggestion. We have made an improvement to Figure 1 by increasing its resolution. (Line number: 142)

Q3. Section 2.5: The information about prank2web is necessary. Any previous studies to support this method is reliable? The authors used this method for a critical task in this work but did not give any information to convince readers that the method is robust. 

A3. Thank you for your comment. We have incorporated detailed information on the prank2web interface, along with relevant references, to better support our method. The added lines are 'PrankWeb is an interface to P2Rank, a machine-learning method for predicting ligand binding sites based on the local chemical neighborhood of a ligand. Our proposed pocket scoring approach, PRANK, uses a Random Forests classifier to prioritize putative pockets and significantly improves the accuracy of Fpocket and ConCavity. To enable a standardized and objective comparison of predicted binding sites across various protein models (Default, Default+Conservation, Alphafold, Alphafold+Conservation), we calculate the pocket probability score using a specific formula that takes into account the number of true and false pockets predicted by each model on the HOLO4K calibration dataset. The probability score for a particular raw score x is obtained by dividing the number of true pockets with a raw score of ≤x (Tx) by the sum of Tx and the number of false pockets with a score of ≥x (Fx). By calibrating the pocket scoring approach using the HOLO4K dataset, the resulting probability score ranges from 0 to 1 and accurately reflects the proportion of true binding sites among all predicted sites with a similar raw score for each model. Ultimately, this standardized approach enables a fair comparison and evaluation of the performance of different protein models in predicting binding sites.' (Line numbers: 152-165)

Q4. The resolution of Figure 2 needs to be improved. Also, the authors need to clarify the confidence level of the structures of these predicted active sites from Alphafold2 (make a connection with Figure 1). If the structures are not reliable then these sites should not be used for further studies. 

A4. Thank you for your precious suggestion and comments.

  1. We have improved the resolution of Figure 2 in the revised manuscript. (Line number: 181)
  2. The probability scores for NnV-Mlp types 1, 2, 3, and 4 were 0.512, 0.583, 0.630, and 0.782, respectively. For more detailed information on the predicted ligand binding sites, please refer to Table S2 in the supplementary file. (Line numbers: 167-169)

Q5. Section 2.6, line 154: RMSD of what? And what is the reference in the calculation?

A5. Thank you for your comment.

  1. We have revised it accordingly as 'The lowest RMSD values for the ligand were selected from all conformations generated during the docking analysis.' (Line number: 193-195)
  2. Additional references for calculating RMSD can be found in section 4.7 (Molecular Docking) of the Materials and Methods. 'Molecular docking is a computational methodology that uses geometric and scoring analyses to predict the interactions between a protein and small molecules. Autodock Vina is a superior docking software compared to Autodock 4.0, due to its higher mean precision in predicting binding modes, faster speed, and automatic pre-calculation of grid maps that are performed internally.' [Reference No. 63,64,]. (Line number: 596-600)

Q6. Section 2.6: Based on the authors' description, it seems the authors considered the docking scores as the binding free energies. This is an incorrect interpretation of docking scores. The docking score can be used for ranking docked poses but does not indicate how tight the compound can bind to the receptor (see https://doi.org/10.1002/cmdc.202200425 and https://doi.org/10.1021/jm050362n). The authors should clarify this in the manuscript and highlight this fact with these previous studies to make sure readers do not misuse docking scores in their own research. In conclusion, docking is not an ideal tool to predict binding free energies.

A6. Thank you for your valuable suggestion. Accordingly, we have changed the “binding affinity” to “binding energy” in the revised version of the manuscript (Section 2.6. Molecular Docking).

Q7. Table 3,4,5: the authors need to ensure the significant figures are consistent (as well as the numbers reported in the main text).

A7. Thank you for your comment. We have changed it appropriately in the revised manuscript.

Q8. Figure 3: the authors need to color the compound based on the atom type (C, N, O,H, etc.). Now it is not easy to see what atoms are forming interactions with protein atoms.

A8. Thank you for your valuable suggestion. We have changed the color of the compound based on the atom type in Figure 3 according to your comment. (Line numbers: 229-232)

Q9. Section 2.9: I'm a little surprised to see that the authors only performed 20 ns simulations (single trajectory for each system) for such a size system. In Figure 5b it is clear that the simulation is not converged (large RMSD change close to the end of the simulation). Also, the RMSD changes in Figure a,b,c are so large in simulations, making me doubt about the docked poses. If the docked poses are not stable, then longer simulations are needed for more reliable analysis. I suggest the authors perform longer simulations (> 200ns). I also notice the authors used PME for electrostatic interactions in simulations. I suggest the authors can try Reaction Field (RF) since previous studies showed that RF can achieve a similar accuracy as PME while RF is much more efficient than PME (https://pubs.acs.org/doi/abs/10.1021/acs.jcim.0c01424, https://onlinelibrary.wiley.com/doi/10.1002/jcc.20828)). If RF is not available in OpenMM then maybe the authors can mention RF in the manuscript so that more readers are aware of this method and it can be used for either longer or cheaper simulations in future work.

A9. Thank you for your valuable comments.

  1. As you mentioned, Figures 5a, 5b, and 5c (corresponding to NnV-Mlp types 1, 2, and 3) exhibit larger RMSD changes due to the presence of more loops in their modeled secondary structures than in Figure 5d (NnV-Mlp type 4). Consequently, the RMSD range for NnV-Mlp types 1, 2, and 3 is greater than that of NnV-Mlp type 4.
  2. We have conducted an RMSD analysis for the ligand, and the docked poses were found to be stable for all four protein types. The results are now included in the revised manuscript in Figures S5a and S5b.
  3. At present, we do not possess adequate computational resources to carry out the 200ns simulation. However, we are currently engaged in identifying potent inhibitors for various types of NnV metalloproteinases that are potentially responsible for causing adverse damage.
  4. We have incorporated your suggestion and revised the manuscript to include the following information: 'In order to identify electrostatic interactions in OpenMM, we utilized Particle Mesh Ewald (PME), as Reaction Field (RF) is not an option in OpenMM.' (Line numbers: 630-632)

Q10. Table 6: The authors need to add units for these numbers.

A10. Thank you for your comment. We have added the units for the MMPBSA binding energy in Table 6. (Line number: 450 and 452)

Q11. Section 2.10: why the authors stated they calculated relative binding energies whereas the absolute binding free energies are reported here (and in Table 6)? Also, since MMPBSA highly depends on the MD simulation. Given the convergence issue I mentioned above, I don't know how much I can trust these numbers. 

A11. Thank you for your kind comments.

  1. We have corrected it as “binding free energy” in section 2.10 and Table 6. (Line numbers: 403 and 457)
  2. The binding free energy of the ligand was calculated using MMPBSA based on the docked poses and dynamic simulations, as shown in Figures S5a and S5b.

Q12. Since the authors failed to provide any experimental data, it is important to make sure the computational modeling is performed with careful designs and rigorous methods. This is the main reason for my comments above.

A12. Thank you for your valuable comment.

  1. We designed our computational modeling using AlphaFold, which is known for its high accuracy in predicting the 3D structure of proteins from their amino acid sequences. The protein models were validated using Ramachandran Plot analysis, and the QMEAN scoring function was applied to ensure the quality of the structures.
  2. We aim to complement our computational findings with experimental data in future studies.

Reviewer 3 Report

In this paper, the authors reported in silico screening of flavonoids as potential inhibitors against metalloproteinase-like proteins in jellyfish Nemopilema nomurai. 3D structures for four types of those proteins were predicted by AlphaFold2, and 39 flavonoids were screened against these structures by molecular docking; 7 of which were further tested for chemical and pharmacological properties using DFT calculation and ADMET prediction. Finally, silymarin was chosen as final candidate and its binding properties were investigated using molecular dynamics simulation.

As poisoning by jellyfish is a common issue, this study is of potential importance for drug discovery. However, the current manuscript contains many errors and weak points that must be corrected before publication, as listed below (though not exhaustively).

- Line 164 and Table 3:
  Residue 296 is duplicated (it seems His, not Arg)

- Figure 3:
  Color coding for panel (d) is inconsistent in the caption.
  Some labels are unreadable.

- Lines 214-216:
  The electronegativity seems inconsistent with Table 4.
  Please also check other values e.g. chemical hardness.

- Sections 2.9 and 4.10
  It seems that only one MD simulation trial was done for each structure.
  As fluctuations are large (including transitions in Fig. 6(c) around 11 ns), conformational sampling may be not enough, and at least three trials are strongly recommended.

- Line 250:
  The reference for the RMSD is not shown.
  (In section 4.10, it is stated that the system was equilibrated for 1+5 ns before the production run. So I wonder which point was taken as the reference.)

- Line 261:
  "The Rg values of the protein and its corresponding ligand complex"
  Does "the protein" mean the protein part of the simulated complex?
  Or have you also tried MD simulation of the protein alone? (in which case please include the result)

- Lines 278-279 and 285:
  These typical values and average range seem inconsistent with Fig. 5(c)(d).
  Could you explain how these values were obtained?

- Figure 5:
  The caption is almost incomprehensible. Could you clearly explain sub-panels (i) to (iv)?
  (Panels (a) to (d) correcpond to types 1 to 4, respectively?)
  For "distance of selected residues", selected residues are not shown, also in the text.

- Table 6:
  Is the unit kcal/mol?
  I think it is meaningless to show to the order of 0.0001 for this case (also Table 4).

- References:
  The references themselves are OK.
  However, there are some strange errors, e.g. author names of refs. 6, 9, and 12; and some page/article numbers are missing (perhaps induced by software like Endnote; please check).

Minor points

- Line 232:
  Table 4 should read Table 5.
  Many table and figure numbers thereafter are also incorrect or missing.

- Table 5:
  Some notations are not correctly compiled; there remain codes e.g. <.

- Line 435:
  Did you also use Generalized Born in this study? (otherwise please delete as it is confusing.)

There are a number of typos, including technical terms; e.g. Position-Boltzmann should read Poisson-Boltzmann. Please check thoroughly.

Author Response

Answers to Reviewers' comments

Q1: Line 164 and Table 3: - Residue 296 is duplicated (it seems His, not Arg)

A1: Thank you for your comments. We have included the information in the revised version of the manuscript. (Line number: 206 and Table 3)

Q2: Figure 3: Color coding for panel (d) is inconsistent in the caption. Some labels are unreadable.

A2: Thank you for your valuable comments. We have revised the caption for Figure 3. (Line number: 237-245)

Q3: Lines 214-216: The electronegativity seems inconsistent with Table 4. Please also check other values e.g., chemical hardness.

A3: Thank you for your comments. We have included the following information in the revised version of the manuscript:

"The results showed that Silymarin (0.16372 eV), Tricetin (0.15419 eV), and Pinobanksin (0.17128 eV) are soft molecules due to their low energy gap values. Tricetin (-0.0744 eV) exhibited the highest HOMO, indicating the highest electron donor, while Apigenin (-0.0638 eV) had the lowest LUMO, indicating the highest electron acceptor. Silymarin (-2061.506039 eV) showed a larger electronegativity value, indicating its inhibitory effect. Eriodictyol (4.295892 eV) had the highest dipole moment followed by Quercetin (4.1933281 eV) and Silymarin (3.991373 eV), which is directly proportional to chemical reactivity. Quercetin (0.07932 eV), Luteolin (0.08467 eV), and Silymarin (0.096935 eV) had lower chemical hardness values, indicating good stability." (Line numbers: 260-268)

Q4 : Sections 2.9 and 4.10 It seems that only one MD simulation trial was done for each structure. As fluctuations are large (including transitions in Fig. 6(c) around 11 ns), conformational sampling maybe not enough, and at least three trials are strongly recommended.

A4: Thank you for your valuable suggestion. As we have performed the MD simulation with limited computing resources, we are unable to conduct another three trials of MD simulation using our facilities. Unlike other research works, our study extensively targeted potent inhibitors for multiple types of jellyfish venom metalloproteinases. This computational study represents primary research, screening active compounds for further work. Our future study will aim to identify active plant metabolites against jellyfish venom metalloproteinases in vitro and in vivo.

Q5 : The reference for the RMSD is not shown.(In section 4.10, it is stated that the system was equilibrated for 1+5 ns before the production run. So, I wonder which point was taken as the reference.)

A5: Thank you for your comments. We have made the following edits to the manuscript based on your suggestions:

  1. "The lowest RMSD values for the ligand were selected from among all conformations generated during the docking analysis." (Line numbers:193-195)

  1. “The preparation protocol consisted of solvent geometry minimization, whole system minimization, and equilibration for 5 ns in the NVT ensemble. The system was gradually heated to 300 K during this equilibration while a weak harmonic potential restraint was placed on the solute. Subsequently, equilibration was carried out in the NPT ensemble while maintaining a weak harmonic restraint on the solute until the desired density was achieved. Finally, a 20 ns production simulation was conducted in the NPT ensemble at 1 atm and 300 K.” (Line numbers: 635-639)

Q6: The Rg values of the protein and its corresponding ligand complex"  Does "the protein" mean the protein part of the simulated complex? Or have you also tried MD simulation of the protein alone? (In which case please include the result)

A6. Thank you for your valuable comment. Based on your suggestion, we have made changes to Figure 5. (Line numbers: 359-363)

Q7: Lines 278-279 and 285: These typical values and average range seem inconsistent with Fig. 5(c)(d). Could you explain how these values were obtained?

A7: Thank you for your comments. We have carefully corrected the values in section 2.9 based on your suggestion. These values were calculated based on the average RMSD in distance.

Q8: Figure 5:  The caption is almost incomprehensible. Could you clearly explain sub-panels (i) to (iv)? (Panels (a) to (d) correspond to types 1 to 4, respectively?) For "distance of selected residues", selected residues are not shown, also in the text.

A8. Thank you for the comments.

  1. We have carefully corrected the caption of Figure 5. (Line Numbers: 398-401)
  2. In section 2.10, we have provided information on the distance between selected residues, and Supplementary Figure S4 shows the corresponding figures.

Q7:Table 6:Is the unit kcal/mol? think it is meaningless to show to the order of 0.0001 for this case (also Table 4).

A7: Thank you for your comments. Based on your suggestion, we have added units to the values presented in Table 6. (Line Number: 450 and 452)

Q8: References: The references themselves are OK However, there are some strange errors, e.g. author names of refs. 6, 9, and 12; and some page/article numbers are missing (perhaps induced by software like Endnote; please check).

A8 : Thank you for your comments. We have revised the reference according to the format specified by the journal. (Line numbers: 724, 732, and 740)

Minor points

Q1- Line 232: Table 4 should read Table 5. Many table and figure numbers thereafter are also incorrect or missing.

A1: Thank you for your comment. We have carefully changed it according to your suggestion.

Q2: Table 5: Some notations are not correctly compiled; there remain codes e.g. <.

A1: Thank you for your comment. We have changed it according to your suggestion. (Line number: 298)

Q3: Did you also use Generalized Born in this study? (Otherwise please delete as it is confusing.)

A3: Thank you for your comment. We have changed it according to your suggestion. (Line number: 519)

Q4: Comments on the Quality of English Language. There are a number of typos, including technical terms; e.g. Position-Boltzmann should read Poisson-Boltzmann. Please check thoroughly.

A4: Thank you for your valuable comment. Based on your suggestion, we have carefully edited and thoroughly checked the revised version of the manuscript for any typos.

Round 2

Reviewer 2 Report

I don't think the authors fully understand my comments about docking scores. Docking scores cannot be considered as predictions of bining affinities or binding energies. As I mentioned in my comment, no correlation has been found between docking scores and binding affinities/energies. This is a well known limitation of docking. In real drug discovery, no one uses just docking scores to select compounds. Much more information/calculation is included in considering potential hits. I am okay if the authors decide to use docking scores to select compounds for their studies. But since no experimental data is given in this work, it is important to highlight the limitation of using docking scores to predict binding affinities/energies for compound selection which is what the authors did in this work. Again, please discuss the limitation of docking for this kind of task since many studies have been done to show this (as I mentioned in my original comments). Otherwise the authors must show experimental data to support these predictions are reliable. 

It is also strange that the authors mentioned RF is not an option in openmm without explaining why they consider RF in this case. Is that because of the studies I mentioned in my comment that show RF is a good alternative for PME? If so, the authors need to clarify it in the manuscript. Besides, RF is implemented in openmm: http://docs.openmm.org/7.1.0/api-c++/generated/OpenMM.NonbondedForce.html It is okay if the authors do not know the best way to use RF in openmm since PME is more commonly used and more tutorials are available. 

Author Response

Reviewer 2

Q1: I don't think the authors fully understand my comments about docking scores. Docking scores cannot be considered predictions of binding affinities or binding energies. As I mentioned in my comment, no correlation has been found between docking scores and binding affinities/energies. This is a well-known limitation of docking. In real drug discovery, no one uses just docking scores to select compounds. Much more information/calculation is included in considering potential hits. I am okay if the authors decide to use docking scores to select compounds for their studies. But since no experimental data is given in this work, it is important to highlight the limitation of using docking scores to predict binding affinities/energies for compound selection which is what the authors did in this work. Again, please discuss the limitation of docking for this kind of task since many studies have been done to show this (as I mentioned in my original comments). Otherwise, the authors must show experimental data to support that these predictions are reliable.

A1: Thank you for your valuable comments.

  1. The experimental data in this work is not feasible due to the following reasons: i) the complexity of purifying a single jellyfish venom protein makes it difficult, and ii) there is currently no recombinant venom protein available from Nemopilema nomurai.
  2. The revised version of the manuscript includes the limitation of docking, as follows:

"Docking algorithms are widely used for their ability to quickly generate poses and scores, but their accuracy is limited by various approximations such as scoring function accuracy, conformational sampling, ligand flexibility, receptor flexibility, and solvent effects. Despite their limitations, these algorithms remain popular because they are user-friendly. The findings of this study provide a micro-level exploration of the inhibitory activity of silymarin against NnV-Mlp types of metalloproteinases. However, further in vitro or in vivo studies are necessary to confirm and validate these results. Our future plan is to express the recombinant jellyfish venom metalloproteinase to identify potential inhibitors, such as silymarin."

 (Line numbers: 678-686)

Q2: It is also strange that the authors mentioned RF is not an option in openmm without explaining why they consider RF in this case. Is that because of the studies I mentioned in my comment that show RF is a good alternative for PME? If so, the authors need to clarify it in the manuscript. Besides, RF is implemented in openmm:http://docs.openmm.org/7.1.0/apic++/generated/OpenMM.NonbondedForce.html It is okay if the authors do not know the best way to use RF in openmm since PME is more commonly used and more tutorials are available.

A2. Thank you for your valuable suggestion. I agree with your comment regarding the effectiveness of RF as an alternative to PME. However, we were not aware that RF is already implemented in openmm, as it has been less commonly utilized than PME. We greatly appreciate the important comments and will consider utilizing RF analysis in our future research projects. (Line numbers: 638-639)

Reviewer 3 Report

All of my previous questions have been answered and, as an initial screening study, the contents are now acceptable.

There are still some typos in the text and strange abbreviations in the reference list (e.g. ref. 8; induced by the software?); which could be corrected in proof, though.

Author Response

Reviewer 3

Q1. All of my previous questions have been answered and, as an initial screening study, the contents are now acceptable.

A1.       Thank you for your kind response.

Q2. Comments on the Quality of English Language: There are still some typos in the text and strange abbreviations in the reference list (e.g. ref. 8; induced by the software?); which could be corrected in proof, though.

A2.       We have carefully revised the manuscript based on your suggestions, regarding the reference format, typographical errors, abbreviation usage, and making English language corrections.
